# Your face scares me: Effects of Perceptual load and Social Anxiety on processing of threatening and neutral faces

**Marios Theodorou**[1]*, **Nikos Konstantinou**[2], **Georgia Panayiotou**[1]

**1** Center for Applied Neuroscience, University of Cyprus, Nicosia, Cyprus, **2** Department of Rehabilitation Sciences, Cyprus University of Technology, Limassol, Cyprus

* theodorou.marios@ucy.ac.cy

## Abstract

Social Anxiety Disorder is among the most widely studied psychiatric conditions. However, the role of attentional and emotional processes in the maintenance of the condition is still not well-established. This study addressed the impact of individual differences in Social Anxiety, by examining the effects of perceptual load and stimulus valence when processing faces vs objects, here used as distractors, within a letter-search task. In addition to RT and accuracy on the letter search task, heart rate, and skin conductance during the task and participants' self-report emotional evaluation were assessed to help interpret performance effects. Results suggest that distractor stimuli that are either threatening or faces impair performance of high SA participants. Results demonstrate a hypervigilance for threatening faces in SA but indicate that this happens primarily when cognitive resources are available, that is, under low perceptual load.

## Introduction

Social Anxiety Disorder (SAD), the clinical form of Social Anxiety (SA) is among the most widely studied psychiatric conditions. Information processing models of the etiology and maintenance of this condition generally suggest that social stimuli, including threatening facial expressions are anxiety-inducing for individuals with SA, activating excessive attentional vigilance and defensive mobilization [1–3].

From an evolutionary perspective, human faces, especially those representing threat (e.g. showing anger), are prominent stimuli for humans, with relevance to survival, as they may signal rejection by conspecifics. Öhman et al. [4–6], hypothesize a common emotional, cognitive and threat related neural pathway, involving brain areas such as the amygdala, for processing faces and other evolutionarily significant fearful stimuli like snakes. Because of their significance, such stimuli are prioritized and instigate automatic, bottom up processing [5, 7]. With regards to SA, Öhman et al.'s model [4, 6], suggests that genetic predisposition and prior experiences of submissiveness during social status conflicts have facilitated fear conditioning to angry faces in high SA individuals, producing hypervigilance towards them.

**Data Availability Statement:** The data underlying this study contain sensitive participant information and cannot be shared publicly. Data access queries

should be directed to the Center for Applied Neuroscience, University of Cyprus (can@ucy.ac. cy).

**Funding:** The author(s) received no specific funding for this work.

**Competing interests:** The authors have declared that no competing interests exist.

Based on Lang's [8] bio-informational theory, semantic memories of emotional stimuli are associated with specific somatovisceral response patterns for survival-related actions. Attention processes share a common neural substrate with emotion, and therefore attentional orienting towards threatening stimuli engages motivational circuits to prepare the organism for action [9]. In anxiety disorders, including SAD, these adaptive orienting and defensive preparation processes are extreme and activated even by mildly threatening or ambiguous stimuli [10–12]. Accordingly, models of SA [13], suggest that socially anxious individuals orient attention to perceived threat, which elevates bodily arousal, that in turn may verify, through conditioning, their fearful perceptions for social situations.

Existing data support, in their majority, the presence of attentional hypervigilance to threatening faces in SA, at least during early stages of processing, at exposure durations of up to 500 ms [14–19]. This seems to hold compared to both low anxious individuals and those with high trait anxiety [20]. However, inconsistent findings also exist, with several studies failing to identify attentional vigilance toward threatening faces in SA. For example, Kolassa and Miltner [21], found interference in gender identification when viewing emotional faces, compared to neutral, suggesting hypervigilance, but this was not specific to angry faces or linked to SA levels. Similarly, de Jong, et al. [22] hypothesized that accuracy on a face identification task with rapid face presentation, would be lower on trials where an emotional stimulus preceded the target, due to emotion-induced blindness. An attentional blink effect suggested vigilance to emotional faces, but this occurred irrespective of emotional expression (angry/happy) and SA.

Other studies, in fact, find lower attention allocation to threatening compared to neutral/ non-face stimuli, with these findings typically interpreted as motivated avoidance in SA for threatening social stimuli. Chen, et al. [23], using the dot-probe paradigm, found that individuals with SAD showed greater attentional bias to household objects at 500ms exposure-times compared to faces, regardless of their emotional expression. Using the 'Face in the crowd' task, Juth, et al. [24] found that participants detected happy faces faster than angry or fearful ones, regardless of SA level, indicating avoidance of negative expressions. Reasons for these mixed results remain currently unclear, perpetuating the question as to the circumstances under which hypervigilance to faces occurs in SA. Conditions such as type of task, stimulus presentation duration, SA severity, types of facial expressions examined and control conditions used may need to be considered for this question to be properly addressed.

First, whether processing emotional faces was a primary task, a secondary task or distractor may modulate attention allocation to face stimuli. Specifically, it has been shown that the processing load that a task imposes affects the processing of task-relevant and irrelevant information [25–28]. According to Load Theory [29, 30] distractibility by task irrelevant stimuli is eliminated due to exhaustion of attentional resources, when the primary task is demanding, i.e. is characterized by high perceptual load. Conversely, when the primary task involves low perceptual load the remaining perceptual capacity is automatically allocated to task irrelevant stimuli. Even though a minority of studies has indicated that faces may represent a special case [31], attracting attention as distractors even under high load conditions, most results indicate that load effects hold for face stimuli as well [32–35].

However, to the degree that high SA individuals preferentially process even mild threat, prioritizing it over any other task, they may show hypervigilance to threatening face distractors, even under high load primary tasks, unlike low SA participants. Soares et al. [36] found that only high SA participants were less accurate on a high load compared to a low load task with threatening faces as distractors, showing preferential attention to these faces. However, this hypothesis has not been extensively examined, and existing results are mixed. Fox, Yates and Ashwin [37], using a modified flanker task, found no difference between low and high SA groups on interference by fear-conditioned angry faces in high load conditions. Under low

load, however, it was the low SA group that showed higher interference for fear-conditioned faces, an effect interpreted as attentional avoidance in SA.

Secondly, most studies that find attentional vigilance towards threatening faces in SA presented face stimuli briefly. Attention allocation may change over time, as proposed by the vigilance-avoidance hypothesis [15], so that automatic hypervigilance to threat during early pre-conscious stages of processing gives way to avoidance, an emotion regulatory process, once the stimulus becomes more conscious.

Thirdly, characteristics of the stimuli may have contributed to the inconsistent findings of attention biases in SA. There is some evidence that high SA individuals are not specifically vigilant towards angry faces, but may find any type of face, and especially neutral, or ambiguous as threatening [12, 38], raising questions as to the appropriate control condition for threatening face stimuli. Also, it remains to be clarified if biases in SA are indeed specific to faces, or if they generalize to other types of survival threat.

In the present study we aimed to address these questions, by examining the presence of attention bias towards threatening faces in SA during very early stages of processing, under varied perceptual load conditions, and using a balanced design that included face, non-face, threatening and neutral distractors, hoping to further specify the circumstances under which these biases occur.

## The present study

We examine attention biases towards threatening faces in SA, using a design that targets very early processing, and varies stimulus characteristics (threatening vs neutral; face vs object) and primary task perceptual load. It was expected that, to the degree that threatening faces are biologically significant for everyone, but particularly for high SA individuals, these participants would show greater difficulty filtering out task irrelevant stimuli of this nature compared to low SA participants, showing preferential attention towards them even when they are to-be-ignored distractors. This will be apparent as slower RT on the primary task and increased emotional reactivity, physiologically and subjectively, to trials containing threatening faces distractors.

Participants high and low on SA performed a letter search task, similar to that used previously [36, 39], with very short stimulus presentation times of 100 ms, capturing stages when automatic attentional vigilance towards threat should occur. In order to identify if there is preferential attention to threatening faces specifically, we also included neutral faces, and objects, threatening (guns) and neutral (chairs) as distractors. This balanced design helps to decipher if it is stimulus emotionality that attracts attention (negative valence; high arousal) or if there is something particular to faces that renders them into more potent distractors than other stimuli. Diverging from Soares et al. [36], distractor stimuli were presented at the center of the letter display to increase the potency of their effects on the primary task. It was expected a) high SA individuals would be more distracted by the presence of threatening faces as distractors, compared to non-distractor trials and trials with other types of distractors (neutral faces, threatening and neutral objects), as indicated by slower RTs. Accuracy was also examined to identify any speed-accuracy trade-off effects. To the degree that high SA individuals prioritize the processing of threatening faces over any other tasks, it was also expected that b) this group and not the low SA participants might show distraction by threatening faces even under high perceptual load conditions.

In addition to performance measures, we utilized physiological indices of emotion-related arousal, assessing heart rate (HR) and skin conductance level (SCL) during the letter search task. Given prior findings [40–42] that high SA participants exhibit greater autonomic

reactivity to emotional faces than controls, we predicted that c) high SA participants would show higher HR and SCL in blocks with threatening faces than any other blocks, and compared to low SA participants.

## Method

### Participants

A power analysis was conducted using the software package, G*Power 3.1.9 [43] to ensure sufficient statistical power. To achieve power of 0.90 to detect a small effect size (f = .15) with a significance level of a = .05 the estimated total required number of 42 participants. In this study, ninety-six college students participated for extra credit and were screened for SA using the Social Phobia and Anxiety Inventory-23 (SPAI-23). Cronbach's alpha for the SPAI-SP subscale was α = .94. Mean SP score for this sample was slightly higher (M = 23.35, SD = 11.63) than the respective original USA normative (M = 22, SD = 10.05) [44]. The bottom 25% (low SA group), and the top 25% (high SA group) of the distribution for SPAI-23 Social Phobia (SP) subscale was selected for the experiment. The final sample consisted of 46 participants (35 women), aged 18–24 (M = 20.67, SD = 1.38). All had normal or corrected to normal vision and provided written informed consent by signing the informed consent form. The study was approved by the Cyprus National Bioethics Committee (Ref: EEBK/EΠ/2014/31). All procedures performed in studies involving human participants were in accordance with the ethical standards of the institutional and/or national research committee and with the 1964 Helsinki declaration and its later amendments or comparable ethical standards. Written informed consent was obtained from all individual participants included in the study. Course credit was offered as an incentive for participation.

### Measures and materials

SPAI-23: The Social Phobia and Anxiety Inventory-23 [45] is an abbreviated version of the SPAI [46], assessing Social Phobia. It was standardized in the Greek language, showing good psychometric properties with adolescents [47], and college students [48].

A total of 64 images (four categories x 16 images; **Table 1**) were used as distractors. Faces (50% women, 50% angry, 50% neutral expression) were selected from the Karolinska Directed Emotional Faces database [49]. Handguns were selected from the UCSD Vision and Memory lab object database [50], and chairs were created at the National Institute of Mental Health [51]. All stimuli were converted to grayscale to avoid confounding by color or luminosity. Faces were cropped around the face to avoid effects of the context and different hair styles using Adobe Photoshop (Adobe systems, Incorporated, San Jose, CA), so that only the central

**Table 1. Stimuli selected for each category.**

| |
|---|
| **KDEF Threatening Faces:** AF01ANS, AF07ANS, AF16ANS, AF20ANS, AF21ANS, AF23ANS, AF25ANS, AF31ANS, AM03ANS, AM05ANS, AM07ANS, AM08ANS, AM10ANS, AM14ANS, AM17ANS, AM29ANS. |
| **KDEF Neutral Faces:** AF01NES, AF07NES, AF16NES, AF20NES, AF21NES, AF23NES, AF25NES, AF31NES, AM03NES, AM05NES, AM07NES, AM08NES, AM10NES, AM14NES, AM17NES, AM29NES. |
| **UCSD Threatening Objects (handguns):** Colt 1911, COlt Gold Cup, Colt official police, CZ 85, CZ_75, hI sTANDARD, LLAMA 87, norinco NP, Norinco NZ85B, Para Ordnance P 14–45, Pietta 1873 Colt, Ruger 22 45, cal, S&W 4506, Taurus PT92, Uberti Schofield, handgun1. |
| **NIMH Neutral Objects (chairs):** 1_Chair_Original, 2_Chair_Original, 4_Chair_Original, 5_Chair_Original, 6_Chair_Original, 8_Chair_Original, 9_Chair_Original, 10_Chair_Original, 11_Chair_Original, 13_Chair_Original, 14_Chair_Original, 15_Chair_Original, 16_Chair_Original, 17_Chair_Original, 18_Chair_Original, 20_Chair_Original. |

face area was visible. Normative ratings of the selected stimuli on valence, arousal, and dominance dimensions exist from an independent college sample [52].

## Apparatus

The experiment was controlled using the Cogent Toolbox (http://www.vislab.ucl.ac.uk/cogent.php) for Matlab (MathWorks, Inc.) on a PC running Microsoft Windows 7 attached to a 19" monitor (60 Hz refresh rate; resolution 800 × 600). A chin rest ensured an invariable distance of 60cm from the monitor.

Physiological data were collected continuously during the letter search task using BIOPAC MP150 for Windows and AcqKnowledge 3.9.0 software (Biopac Systems Inc., Santa Barbara, CA). Ag/AgCl shielded electrodes were placed on participants' face and arms following standard procedures [53]. Electrocardiogram (ECG) recorded at the two inner forearms was filtered by a BIOPAC ECG100C bioamplifier recording HR between 40 and 140 beats per minute (BPM). SCL was recorded using a BIOPAC GSR100C transducer amplifier.

## Procedure

Participants arrived at the lab, and sat in a dimly lit, sound attenuated room. Following informed consent, they were given instructions and fitted with physiological monitors. A 5-min adjustment period, during which participants were asked to sit quietly and relax, preceded the experiment to stabilize physiological recordings and familiarize them with the equipment. Next, participants completed the letter-search task for which they had to identify which of the two pre-specified target-letters (either X or Z) appeared in a display of other letters. Each trial began with a fixation cross (0.4˚ x 0.6˚) presented at the center of the screen for 1000 ms, followed by presentation of eight letters (0.4˚ x 0.6˚) for 100 ms. Letters were placed with equal probability in any of eight positions arranged in a circle of 2˚ in radius centered at fixation. The task included two load conditions: in the low load condition Os were presented in seven of the letter places (**Fig 1**). In the high load condition, seven letters with similar configuration to the X and Z target-letters were presented (K, F, V, T, L, N, H; **Fig 2**). A distractor (2.5˚ × 2.5˚) was presented in the middle of the circle on 50% of the trials (randomly intermixed), based on evidence that participants tend to habituate to more frequent distractor presentations, which can obliterate distractor effects [35, 39, 54]. A smaller % of trials with distractors (20%) was pilot tested, yielding similar results, therefore a higher % was used to allow for more samples of each distractor type. Participants were instructed to search for a target letter "X" or "Z", and respond as quickly and accurately as possible by pressing the keys 1 or 2 on the numerical keypad for "X" or "Z" respectively, while ignoring task-irrelevant images. The target-letter was equally likely to appear in any of the eight positions of the circle.

Trials varying on distractor emotionality (threatening/neutral) and perceptual load (high/low) were presented in separate counterbalanced blocks. Distractor present and distractor absent trials were mixed randomly within all blocks. Each participant completed 8 blocks of 64 trials each (low load threatening faces, high load threatening faces, low load neutral faces, high load neutral faces, low load threatening objects, high load threatening objects, low load neutral objects, high load neutral objects), in counterbalanced fashion (ABBABAAB). Blocks were separated by 60s rest intervals. Prior to the experiment, participants completed two practice blocks of 16 trials each. After task completion, participants rated their subjective experience while viewing the distractor stimuli on valence (anchors: pleasant; unpleasant), arousal (anchors: intense; calm), and dominance (anchors: dominant; submissive), to verify that stimuli elicited the intended response.

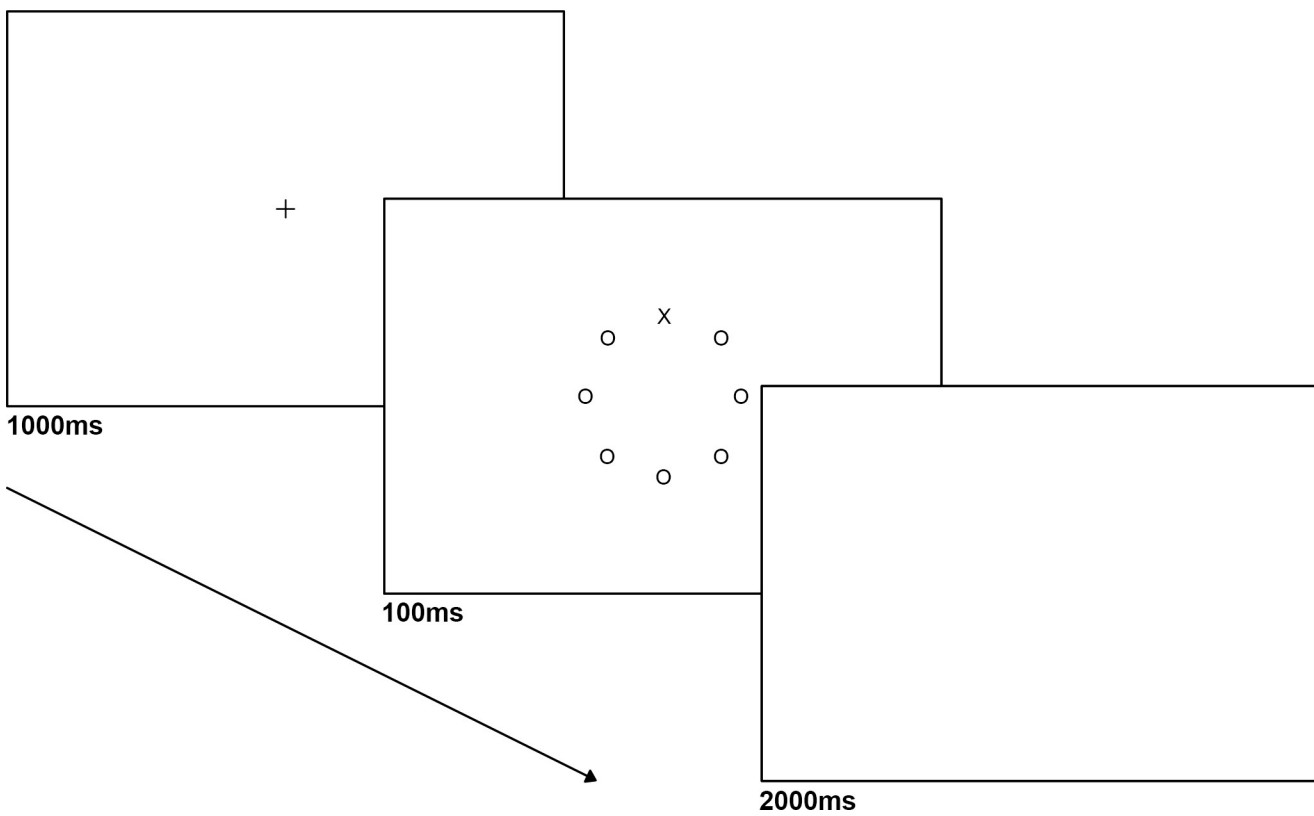

**Fig 1. An example trial sequence and timing in visual search task (low load distractor absent).**

### Data reduction

Trials with RTs of less than 150 ms, more than 1,500 ms, or +/- 2.5SD from each participant's mean, were considered outliers (8%) and were excluded from analyses. Mean accuracy was calculated as the percentage of correct trials per block. Only correct trials in each type of block were used for the calculation of mean RT [39, 54]. For HR and SCL, means were computed for each type of block and across all inter-trial intervals. Three participants (one low SA, two high SA) were removed from HR and SCL analyses due to poor signal. Mean physiology across all inter-trial intervals was the covariate for analyses on physiological measures to account for individual differences.

For performance effects, a repeated measures ANOVA in Statistical Package for the Social Sciences-25.0 examined separately RT and accuracy (% correct) as the dependent variables, with load (low, high), distractor presence (present, absent), distractor emotionality (threatening, neutral), and distractor type (face, object) as within-subject variables, and social anxiety (low, high) as a between subject variable. A similar ANOVA, with load, distractor emotionality, and distractor type, as within-subject variables, and social anxiety as a between subject variable examined effects on HR and SCL. Because these measures were taken as the average of each block across both distractor present and absent trials, distractor presence was not a variable in these analyses. The Benjamini-Hochberg procedure [55] was used to control for multiple testing bias (false positives). To adjust with the low effect sizes in attentional bias research [2], the false discovery rate was set at 0.10 [56, 57]. Effects are reported at a significance level of $p < .05$.

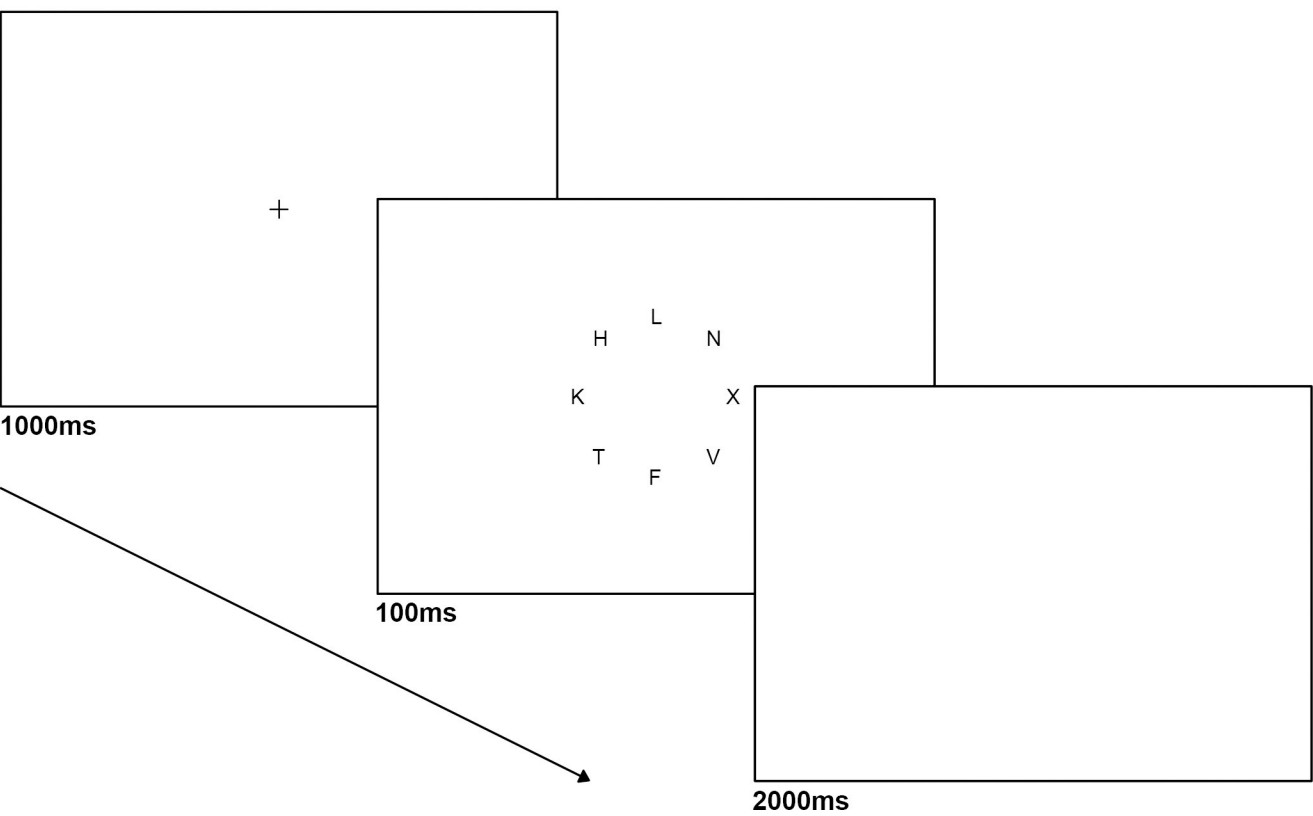

**Fig 2. An example trial sequence and timing in visual search task (high load distractor absent).**

## Results

### Manipulation checks

**Effects of load and distractor presence on performance.** Table 2 presents mean RTs and Accuracy rates in each condition of distractor presence and load. A significant effect of load, $F(1, 44) = 274.42$, $p < .01$, $\eta^2 = .86$, showed that participants were slower in high load ($M = 741.15$ms, $SD = 151.64$) compared to low load blocks ($M = 540.17$ms, $SD = 126.94$). A significant effect of distractor presence was also found, $F(1, 44) = 28.79$, $p < .01$, $\eta^2 = .40$, so that participants were slower in distractor present trials ($M = 649.30$ms, $SD = 141.38$) compared to distractor absent trials ($M = 632.02$ms, $SD = 137.19$). A significant load x distractor presence interaction $F(1, 44) = 9.46$, $p = .004$, $\eta^2 = .18$, indicated that the distractor presence effect was only significant in the low load blocks, $F(1, 44) = 48.30$, $p < .01$, $\eta^2 = .52$, with participants being slower in distractor present trials ($M = 552.98$ms, $SD = 129.22$) compared to distractor absent trials ($M = 527.36$ms, $SD = 124.65$). Distractor type and emotionality did not show any significant effects on RT or an interaction with load.

With regards to accuracy, a significant effect of load, $F(1, 44) = 262.23$, $p < .01$, $\eta^2 = .86$, showed that participants were more accurate in low load ($M = .941$, $SD = .08$) compared to high load trials ($M = .687$, $SD = .14$). A significant effect of distractor presence, $F(1, 44) = 33.38$, $p < .01$, $\eta^2 = .43$, indicated that they were more accurate on distractor absent ($M = .826$, $SD = .10$) compared to distractor present trials ($M = .802$, $SD = .11$). There was no significant interaction between load x distractor emotionality, $F(1, 44) = .76$, $p = .39$, or type, $F(1, 44) = 1.72$, $p = .20$, suggesting that effects of load remain irrespective of these variables.

**Table 2. Mean RTs, accuracy rates, and ratings per distractor category (SD in parentheses).**

| | Low Load | | | | | | | |
|---|---|---|---|---|---|---|---|---|
| | Trials with distractor | | | | Trials without distractor | | | |
| | TF | NF | TO | NO | TF | NF | TO | NO |
| RT ms | 559.2 (133.4) | 545 (13.4) | 558.5 (120.8) | 549.3 (128.4) | 526.6 (113.7) | 525.7 (128.9) | 526.04 (119.97) | 531.12 (135.4) |
| % Acc | 92% | 95% | 92% | 94% | 96% | 94% | 96% | 95% |
| | High Load | | | | | | | |
| | Trials with distractor | | | | Trials without distractor | | | |
| | TF | NF | TO | NO | TF | NF | TO | NO |
| RT ms | 743.9 (140.4) | 742.2 (142.5) | 750.9 (161.8) | 745.5 (169.5) | 740.5 (141.2) | 734.6 (139.9) | 734.7 (160.04) | 736.9 (157.8) |
| % Acc | 68% | 69% | 65% | 67% | 68% | 73% | 71% | 69% |
| | Ratings | | | | | | | |
| | TF | | NF | | TO | | NO | |
| Valence | 3.53 (.89) | | 4.05 (.71) | | 3.11 (1.21) | | 5.23 (1.29) | |
| Arousal | 5.78 (.95) | | 5.38 (.89) | | 6.31 (1.30) | | 4.20 (1.10) | |
| Dominance | 5.79 (1.79) | | 6.10 (2.82) | | 5.68 (2.22) | | 6.74 (1.61) | |

TF = Threatening Faces, NF = Neutral Faces, TO = Threatening Objects, NO = Neutral Objects Ratings: Note. Means indicate the average of the 16 images per category. Likert scales for valence (1 = very unpleasant, 9 = very pleasant), arousal (1 = very relaxed, 9 = very tense), dominance (1 = no control over the situation, 9 = full in control of the situation

**Subjective reactivity.** Table 1 presents mean valence, arousal and dominance ratings for each distractor category. A Repeated Measures ANOVA with distractor emotionality (threatening, neutral), and distractor type (face, object), as within-subject variables, and SA (low, high) as a between subject variable showed no significant main or interactive effects of SA on ratings. With regards to valence, a significant interaction of distractor type x emotionality interaction, $F(1, 42) = 35.20$, $p < .001$, $\eta^2 = .46$, indicated that participants, reported feeling more positive when viewing neutral compared to threatening faces. A significant interaction between distractor type and emotionality, $F(1, 42) = 40.22$, $p < .001$, $\eta^2 = .49$, showed that they also reported less arousal in the presence of neutral compared to threatening faces $F(1, 42) = 12.26$, $p < .0001$, $\eta^2 = .23$, and higher arousal in the presence of threatening compared to neutral objects, $F(1, 42) = 72.23$, $p < .001$, $\eta^2 = .63$. A significant interaction between distractor type and emotionality, $F(1, 42) = 8.06$, $p = .007$, $\eta^2 = .16$ indicated that participants reported less dominance in the presence of threatening compared to neutral faces, and threatening compared to neutral objects $F(1, 42) = 18.60$, $p < .001$, $\eta^2 = .31$.

**Physiological reactivity.** A significant interaction between distractor type and distractor emotionality was found, $F(1, 40) = 4.34$, $p = .044$, $\eta^2 = .10$. This interaction was broken down by examining the effect for each distractor type, and showed that participants exhibited higher HR in blocks with threatening ($M = 79.92$, $SD = 13.69$) compared to neutral faces ($M = 79.72$, $SD = 11.86$), $F(1, 40) = 7.45$, $p = .009$, $\eta^2 = .16$.

## Main hypotheses

**Effects of SA on performance.** Looking at the between group effects from the ANOVA examining RT, a significant interaction between distractor type and SA level was observed, $F(1, 44) = 4.93$, $p = .03$, $\eta^2 = .10$. This was broken down by examining the distractor type effect in each SA group separately. Low SA participants were slower in RT on blocks with distractor objects compared to faces, $F(1, 22) = 4.73$, $p = .041$. Among high SA participants the distractor type effect was non-significant, $F(1, 22) = 1.33$, $p = .26$, though RT was numerically slower in

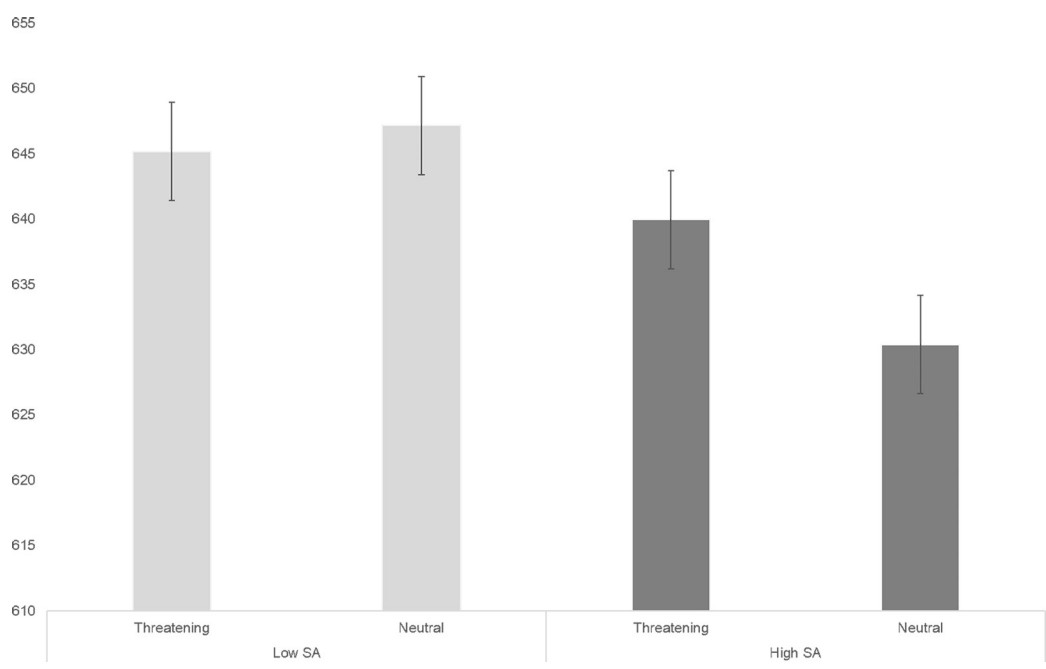

**Fig 3. Interaction between SA level and distractor emotionality (RT).** Error bars represent standard errors.

blocks with distractor faces ($\underline{M}$ = 640.12, $\underline{SD}$ = 122.87) compared to objects ($\underline{M}$ = 630.22, $\underline{SD}$ = 133.23). Additionally, a significant distractor emotionality x SA level interaction was found, $\underline{F}$ (1, 44) = 4.14, $\underline{p}$ = .048, $\eta^2$ = .09. When examining each SA group separately, low SA partici-pants were equally fast in RT regardless of distractor emotionality, $\underline{F}$(1, 22) = .27, $\underline{p}$ = .61, while in the high SA group an effect of distractor emotionality, $\underline{F}$(1, 22) = 5.17, $\underline{p}$ = .033, $\eta^2$ = .19, indicated participants were slower on blocks with threatening, compared to neutral distractors, as hypothesized (**Fig 3**). There were no significant interactions between SA and within-sub-jects variables with regards to accuracy.

**Effects of SA on physiology.** **Table 3** presents mean HR for each distractor category per SA group. There was a significant effect of SA level on HR, $\underline{F}$(1, 40) = 8.94, $\underline{p}$ = .005, $\eta^2$ = .18, so that there was higher overall heart rate in the high SA group ($\underline{M}$ = 81.01, $\underline{SD}$ = 12.11) com-pared to the low SA group ($\underline{M}$ = 78.45, $\underline{SD}$ = 12.78). This was modified by a significant two-way interaction between distractor type and SA level, $\underline{F}$(1, 40) = 3.97, $\underline{p}$ = .05, $\eta^2$ = .09. The interaction was broken down by examining the distractor type effect in each SA group. Dis-tractor type was non-significant, $\underline{F}$(1, 20) = 1.21, $\underline{p}$ = .29, in the low SA group, but significant in

**Table 3. Mean and standard deviation for HR per distractor category in low and high SA groups.**

| | | | Mean | SD |
|---|---|---|---|---|
| Low SA | Threatening | Faces | 77.55 | 12.56 |
| | | Objects | 77.92 | 12.73 |
| | Neutral | Faces | 78.11 | 12.67 |
| | | Objects | 78.75 | 13.16 |
| High SA | Threatening | Faces | 82.40 | 14.72 |
| | | Objects | 81.08 | 11.36 |
| | Neutral | Faces | 81.41 | 11.11 |
| | | Objects | 80.69 | 11.25 |

the high SA group, $\underline{F}(1, 19) = 5.99$, $\underline{p} = .024$, $\eta^2 = .24$ so that in blocks with faces, high SA participants showed higher HR ($\underline{M} = 81.51$, $\underline{SD} = 12.91$), compared to blocks with objects ($\underline{M} = 80.51$, $\underline{SD} = 11.30$). A significant two-way interaction between distractor emotionality and SA level was also obtained, $\underline{F}(1, 40) = 5.70$, $\underline{p} = .022$, $\eta^2 = .13$. For the low SA group there was no significant effect of distractor emotionality $\underline{F}(1, 20) = .004$, $\underline{p} = .95$; for high SA participants HR was higher in blocks with threatening distractors ($\underline{M} = 81.31$, $\underline{SD} = 13.04$) compared to neutral [$\underline{M} = 80.68$, $\underline{SD} = 11.18$; $\underline{F}(1, 19) = 18.55$, $\underline{p} < .001$, $\eta^2 = .49$]. The 3-way SA x type x emotionality interaction was not significant, however, looking at the means in Table 2, suggests that the highest HR was for blocks with threatening faces. A non-significant trend for a three-way load x emotionality x SA interaction indicated additionally that for high SA participants the increased HR for threatening relative to neutral distractors only occurred under low perceptual load conditions $\underline{F}(1, 19) = 3.36$, $\underline{p} = .074$, $\eta^2 = .08$. No significant effects of SA or interactions were found for SCL.

## Discussion

This study examined attention biases in SA, taking into consideration task perceptual load, stimulus type and valence, using a task focusing on early processing stages where vigilance is mostly expected. We used multiple dependent measures, including RT, psychophysiology and self-report to clarify the circumstances under which vigilance occurs. The study contributes three main findings, that converge to partially support our hypothesis that high SA individuals show increased vigilance towards threatening faces during early stages of processing. Findings additionally contribute useful insights about the specific circumstances during which such effects appear.

First, high SA participants showed increased distractibility by faces, evidenced in their RT. While low SA participants were more distracted by objects compared to faces, in the high SA group interference was equivalent for both objects and faces, showing that the significance of faces was greater for this group, in accord with our hypothesis. Looking at the means, one can observe that whereas for low SA participants distraction was mostly carried by threatening objects (i.e. guns), high SA participants found both faces and guns to be equally distracting, with threatening faces resulting in the slowest RTs albeit non-significantly so. Thus, threatening faces, in accord with our hypothesis and existing theories [4–6], hold special significance in SA. The very short stimulus presentation used may have permitted for early attentional vigilance, leading to slower RTs on the primary task to become apparent. We did not obtain the expected type x emotionality interaction in the high SA group, which would have documented more definitively that threatening faces attract significantly more attention than threatening objects, in spite of means being in the expected direction; this may be due to the particular type of threatening objects included: Guns may have signalled immediate survival threat for both high and low SA participants alike, evoking high levels of vigilance and preventing this difference from emerging more clearly. The special significance of threatening faces for high SA participants, in accord with our main hypothesis, is however, also corroborated by the increased HR on blocks with face distractors for this group only, especially threatening ones, which suggests that faces elicit defensive action preparation but only for those socially anxious. The reason that psychophysiological effects were seen on HR but not SCL may be due to the fact that HR is sensitive to both valence and arousal aspects of emotion [58], whereas SCL is more sensitive to arousal. What makes a threatening face elicit defensive action may entail both its arousal and aversiveness value.

Secondly, high SA individuals had slower RTs on trials with threatening than neutral distractors, showing that their attention was attracted by potentially survival related stimuli,

irrespective of their type. Although this may be an appropriate response if the threat is real or significant, low SA participants did not show this general effect, remaining apparently focused on the primary task, as instructed irrespective of the emotionality of distractor stimuli. In contrast, even in the safe context of the lab, with no real danger present, socially anxious individuals were unable to filter out threatening distractors of any type, and showed slowing in their performance and defensive preparation; this is in accordance with [4, 5], Lang, and others who describe anxious individuals as having exaggerated defensive reactions even at low levels of threat.

Third, there was no significant interaction between stimulus type or emotionality with Load for either group. This suggests that perceptual load effects on RT held irrespective of type of distractor stimulus for both high and low SA participants, which is in contrast to our hypothesis that faces would hold such high significance for high SA participants as to override load effects and occur even under high perceptual load. As indicated by multiple prior studies [33–35], the high perceptual load task seemed to absorb available perceptual resources, prohibiting any significant levels of distraction, at least on RT measures. The marginal interaction showing that increased HR to threating stimuli in the high SA group occurred only under low perceptual load, concurs with this as well, as defensive mobilization seems to have taken place to a greater degree under conditions where threatening distractors could be more fully perceived, irrespective of how important they were to the individual. Alternatively, one may wonder whether failure to support this hypothesis may suggest that our high SA participants were not anxious enough to demonstrate extreme threat sensitivity. Although participants were non-clinical, the top quartile were selected as high SA were actually higher in SA compared to the SPAI-23 USA college sample, suggesting that significant levels of anxiety were present. Future studies should however, replicate these effects on participants with clinical levels of SAD to validate the present conclusions. Another possibility is that our face stimuli were not threatening enough for the anxious participants, as might be suspected by the absence of SA level x type interaction on subjective ratings. However, the HR and RT effects argue against this possibility and show that threatening faces were perceived as significant and arousing by our high SA group.

Credibility of these results is enhanced by the fact all our manipulations appeared to be effective. Analyses indicated that perceptual load was appropriately manipulated and produced expected effects. Participants were overall slower and less accurate on blocks where the task involved high rather than low perceptual load and when distractors were present. RT was negatively affected by distraction only on low load blocks, consistent with Load Theory [31, 32, 39, 54].

Distractor stimulus emotionality was also manipulated effectively, as shown by subjective and HR responses: Participants rated threatening stimuli, especially faces, as more negative, arousing and leading to less perceived dominance than neutral. Threatening stimuli, particularly faces, also led to increased HR relative to neutral faces and objects, concurring with the idea that such stimuli have significance to human beings and elicit defensive activation [40–42].

Results of this study should be seen in light of its strengths and limitations. Among its strengths is the use of multiple measures of engagement with the stimuli, including performance and physiological indices. Our findings converge to demonstrate hypervigilance to threatening stimuli, especially threatening faces, for high SA individuals, as expected. The short time frames of stimulus presentation allowed us to capture vigilance effects during early processing, when they should be most apparent. Moreover, the balanced design and multiple manipulation checks permitted us to examine effects of threatening faces versus other emotional and neutral stimuli on attention and show that high SA individuals are more attentive to

threat overall, but are most sensitive to threatening faces. Limitations include the non-clinical nature of the sample, which raises the need for replication among individuals with clinical levels of SA. Additionally, the fact that guns represented the threatening objects may have obliterated more specific attention biases to threatening faces, because guns may signal immediate survival threat and may be equally relevant for both participant groups, eliciting high levels of engagement. Using alternative aversive object stimuli in the future may extend and corroborate present findings. Another limitation pertains to the inability to examine gender effects, due to the primarily female nature of the student body at the University where the sample was recruited. Future research could use gender-balanced and larger community samples, reflecting a normative population. This research can also be extended to examine SA biases towards faces depicting for example disgust or fear.

In spite of some limitations, findings of this study contribute insights, supporting vigilance for threatening faces in SA at very short exposure durations, but indicating that this happens primarily when cognitive resources are available, that is under low perceptual load. It is also consistent with emotion theories [8, 9] and clinical models [13], regarding the cognitive prioritization of stimuli that threaten survival and the distinctive emotional mobilization for personally threatening stimuli in SA.

## Author Contributions

**Conceptualization:** Marios Theodorou, Georgia Panayiotou.

**Data curation:** Marios Theodorou.

**Formal analysis:** Marios Theodorou.

**Methodology:** Marios Theodorou, Nikos Konstantinou, Georgia Panayiotou.

**Project administration:** Marios Theodorou.

**Resources:** Georgia Panayiotou.

**Software:** Nikos Konstantinou.

**Supervision:** Nikos Konstantinou, Georgia Panayiotou.

**Writing – original draft:** Marios Theodorou.

**Writing – review & editing:** Marios Theodorou, Georgia Panayiotou.

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
