## [Decision Letter · Decision Letter 0]

4 Mar 2021

Your face scares me: Effects of Perceptual load and Social Anxiety on processing of threatening and neutral faces

PONE-D-20-37708

Dear Dr. Theodorou,

We’re pleased to inform you that your manuscript has been judged scientifically suitable for publication and will be formally accepted for publication once it meets all outstanding technical requirements.

Kind regards,

Christos Papadelis, Ph.D.

Academic Editor

PLOS ONE

Journal Requirements:

1. In your Methods section, please ensure you have specifically stated whether you obtained consent from parents or guardians of the minors included in the study or whether the research ethics committee or IRB specifically waived the need for their consent.

4. We note that Figures 1 and 2 include an image of a participant in the study. 

As per the PLOS ONE policy (http://journals.plos.org/plosone/s/submission-guidelines#loc-human-subjects-research) on papers that include identifying, or potentially identifying, information, the individual(s) or parent(s)/guardian(s) must be informed of the terms of the PLOS open-access (CC-BY) license and provide specific permission for publication of these details under the terms of this license.

Please download the Consent Form for Publication in a PLOS Journal (http://journals.plos.org/plosone/s/file?id=8ce6/plos-consent-form-english.pdf). The signed consent form should not be submitted with the manuscript, but should be securely filed in the individual's case notes.

Please amend the methods section and ethics statement of the manuscript to explicitly state that the patient/participant has provided consent for publication: “The individual in this manuscript has given written informed consent (as outlined in PLOS consent form) to publish these case details”.

Please respond by return e-mail with an amended manuscript. We can upload this to your submission on your behalf.

If you are unable to obtain consent from the subject of the photograph, please either instruct us to remove the figure or supply a replacement figure by return e-mail for which you hold the relevant copyright permissions and subject consents. In some cases, you may need to specify in the text that the image used in the figure is not the original image used in the study, but a similar image used for illustrative purposes only. We can make any changes on your behalf.

Reviewers' comments:

Reviewer's Responses to Questions

**Comments to the Author**

1. Is the manuscript technically sound, and do the data support the conclusions?

Reviewer #1: Yes

Reviewer #2: Yes

2. Has the statistical analysis been performed appropriately and rigorously? 

Reviewer #1: Yes

Reviewer #2: Yes

3. Have the authors made all data underlying the findings in their manuscript fully available?

Reviewer #1: Yes

Reviewer #2: Yes

4. Is the manuscript presented in an intelligible fashion and written in standard English?

Reviewer #1: Yes

Reviewer #2: Yes

5. Review Comments to the Author

Reviewer #1: The authors examine how individuals high in social anxiety, compared to those low in social anxiety, can more readily be distracted by threatening stimuli, especially faces. They also explore the role of perceptual oad. The study was well-designed, well-executed. It was also notable that the authors collected physiological measures as well. Most of the hypotheses were supported, and the authors had reasonable explanations for hypotheses that were not supported (e.g., why threatening faces were not significantly different from threatening objects for those high in SA; why no interaction with stimuli type and emotionality with perceptual load); they also used supplementary physiological measures (e.g., HR) to buttress their accounts. All in all, this one-study manuscript has all the strengths of a one-study manuscript (and all the limitations of a one-study manuscript). The authors also present a comprehensive literature review in the introduction which later allows them to be circumspect and contextualize their findings, describing how their findings dovetail with the rest of the literature.

One limitation that the authors do not address is the decision to focus on the bottom and top quartiles of the SA measures. They report that the average of their sample is a little high, compared to other normative samples. So, why not just a median split to get at the difference between hi and low SA? Or why not keep the SA measure as a continuous variable, and search for interactions? Keeping individual differences measures continuous, instead of splitting them, seems more common in personality and social psychology research. Anyhow, that issue would be good to explain.

Reviewer #2: On the whole I think this is a straightforward and cleanly executed study. The effects in question are to my knowledge novel and demonstrate an intriguing relationship between cognitive load and social anxiety. One interesting consideration that could be a useful way to explore the role of the face itself vs. the low-level distracting information in the face pattern could be consider either negating the contrast of the face or presenting the face image upside-down, both of which are known to reduce the efficiency of face recognition substantially. However, I think the results are presented here with appropriate nuance and the current design supports interesting conclusions that may lead to further work in this area.

6. PLOS authors have the option to publish the peer review history of their article (what does this mean?). If published, this will include your full peer review and any attached files.

Reviewer #1: No

Reviewer #2: No

---

## [Editor Report · Acceptance letter]

16 Mar 2021

PONE-D-20-37708 

Your face scares me: Effects of Perceptual load and Social Anxiety on processing of threatening and neutral faces

Dear Dr. Theodorou:

I'm pleased to inform you that your manuscript has been deemed suitable for publication in PLOS ONE. Congratulations! Your manuscript is now with our production department. 

Kind regards, 

on behalf of

Dr. Christos Papadelis 

Academic Editor

PLOS ONE